# Epigenetic Mechanisms of Aging and Aging-Associated Diseases

**DOI:** 10.3390/cells12081163

**Published:** 2023-04-14

**Authors:** Annamaria la Torre, Filomena Lo Vecchio, Antonio Greco

**Affiliations:** 1Laboratory of Gerontology and Geriatrics, Fondazione IRCCS Casa Sollievo della Sofferenza, San Giovanni Rotondo, 71013 Foggia, Italy; 2Complex Unit of Geriatrics, Department of Medical Sciences, Fondazione IRCCS Casa Sollievo della Sofferenza, San Giovanni Rotondo, 71013 Foggia, Italy

**Keywords:** epigenetics, aging, diseases, methylation, histone modifications

## Abstract

Aging is an inevitable outcome of life, characterized by a progressive decline in tissue and organ function. At a molecular level, it is marked by the gradual alterations of biomolecules. Indeed, important changes are observed on the DNA, as well as at a protein level, that are influenced by both genetic and environmental parameters. These molecular changes directly contribute to the development or progression of several human pathologies, including cancer, diabetes, osteoporosis, neurodegenerative disorders and others aging-related diseases. Additionally, they increase the risk of mortality. Therefore, deciphering the hallmarks of aging represents a possibility for identifying potential druggable targets to attenuate the aging process, and then the age-related comorbidities. Given the link between aging, genetic, and epigenetic alterations, and given the reversible nature of epigenetic mechanisms, the precisely understanding of these factors may provide a potential therapeutic approach for age-related decline and disease. In this review, we center on epigenetic regulatory mechanisms and their aging-associated changes, highlighting their inferences in age-associated diseases.

## 1. Introduction

Aging is a multifactorial biological process of declining physiological functions, increasing the susceptibility to aging-related chronic diseases [1]. Indeed, aging is the primary risk factor for major human pathologies including cancer, metabolic, musculoskeletal and cardiovascular disorders, as well as neurodegenerative diseases [2]. Often this increased risk represents the principal cause of death which accompanies advancing age [3]. For those reasons, a great number of studies have centered on the clarification of the hallmarks of aging for identifying potential target therapeutic molecules to attenuate the aging process. 

Nowadays, among the hallmarks of aging are: senescence, stem cell exhaustion, genomic instability, altered intercellular communication and epigenetic deregulation [2]. In general, epigenetics refers to heritable changes in gene expression that are, unlike mutations, not attributable to alterations in the sequence of the DNA [4].

Among the epigenetic events are: DNA and histone modifications, ATP-dependent chromatin remodeling complexes and non-coding RNAs [5,6,7,8]. In this review, we first provide a description of the functioning of direct methylation of the DNA, and the modification of the proteins that package the DNA (the histones), in general, before moving to their modification of and implication in aging-associated diseases.

## 2. Epigenetic Mechanisms

### 2.1. DNA Methylation

DNA methylation represents the most studied and best understood among the epigenetic mechanisms [9]. It occurs directly at the DNA level, regulating gene expression by engaging the proteins required in gene silencing or impeding the interaction between DNA and transcription factors [10]. For instance, DNA methylation is implicated in the repression of retroviral elements, in genomic imprinting and X chromosome inactivation, alternative splicing and the regulation of tissue-specific gene expression [11]. The DNA methyltransferases (DNMTs) are a family of enzymes involved in DNA methylation. They catalyze the transfer of a methyl group (5mC) from S-adenosyl-methionine (SAM) to the carbon-5 position of the cytosine residues in the CpG. Among the DNTM family are: DNMT1, DNMT2, DNMT3A, DNMT3B and DNMT3L. While DNMT1, DNMT3A and DNMT3B are canonical DNMTs that catalyze the addition of methylation marks to genomic DNA, DNMT2 and DNMT3L are non-canonical family members. Indeed, whereas DNMT2 has tRNA methyl transferase activity, DNMT3L does not possess any catalytic DNA or RNA methyl transferase activity [12,13]. The canonical DNMTs preferentially modify cytosine, followed by a guanine residue, defined as CpG islands. Normally they are assembled in large clusters, approximately 1000 base pairs long, at the promoter and within exonic regions [14]. It has been demonstrated that 70–80% of these clusters are methylated in somatic cells [15]; whereas, demethylation of CpG islands occurs often in human embryonic or induced pluripotent stem cells [16]. Other regions of the mammalian genome contain few CpG dinucleotides that are extensively methylated [16]. For instance, CpG methylation of most of the genome ensures genomic stability by controlling repetitive DNA elements, such as interspersed and tandem repeats (LTR: long-terminal repeats, LINE: long interspersed nuclear elements, SINE: short interspersed nuclear elements) [17]. Generally, while DNA methylation regulates the transcriptional initiation activity, the DNA hypomethylation is positively associated with higher transcriptional activity through a multistep process catalyzed by DNA demethylase [18]. This latter process is catalyzed by the ten–eleven translocation (TET), a family of proteins that, during the enzymatic reaction, uses oxygen, Fe(II) and α-ketoglutarate as substrates. The reaction, firstly, consists of the oxidation of 5-mC to 5-hydroxymethylcytosine (5-hmC). Consequently, the TETs continue the oxidation of 5-hmC to 5-formylcytosine (5-fC) and 5-carboxylcytosine (5-caC) [19,20]. Among the TET family are the proteins Tet1, Tet2 and Tet3. Albeit all having the ability to oxidize 5-mC, they are implicated in diverse biological processes, partially ascribable to their differential expression through development and in a cell-type specific manner [21]. Indeed, Tet1 is amply expressed in embryonic stem cells (ESCs), but it is generally downregulated during differentiation, whereas Tet2 and Tet3 exhibit a superimposable expression pattern in various tissues [22]. 

### 2.2. Histone Modifications

DNA methylation is able to regulate gene expression either directly, by inhibiting the link between the binding site and its relative transcription factors, or indirectly by its interaction with proteins, termed methyl-binding proteins (MBPs) [20]. Indeed, once DNA is methylated, DNA methyl-binding proteins (MBPs) are able to link to the DNA for engaging transcriptional corepressors, including polycomb proteins, histone deacetylase (HDAC) complexes and chromatin remodeling complexes [23], which ultimately changes the transcriptional activity of genes. The eukaryotic genome is packaged in a complex structure named chromatin, that regulates the compactness of the genome and conditioning, and therefore the accessibility of regulatory proteins and RNA polymerases to DNA. The fundamental unit of chromatin is the nucleosome, that is formed by an octamer of histones, composed of two copies of each histone (H2A, H2B, H3 and H4, as well as histone variants, such as macroH2A, H3.3 and H2A.Z), surrounded by 147 base pairs of DNA [24]. As the DNA moves in and out of its 1.65 turns around the nucleosome, a histone linker, H1, binds to it. Nucleosomes are subject to post-translationally changeable acting on histone N-terminal tails [25]. The modifications include methylation, acetylation, phosphorylation, ribosylation, ubiquitination and sumoylation [26,27]. These modifications are reversible and, together with the presence of different isoforms, permit the existence of different spatial rearrangements of chromatin that influence the interaction of DNA–RNA polymerase II and transcription factors. 

### 2.3. Histone Methylation

Histone methylation acquires a dual function in the regulation of gene expression. Indeed, it is involved in the activation or silencing of gene transcription, depending on the number of methyl groups bound and the specific amino acid residue [28]. It is achieved at the residues of lysine (K) and arginine (R) by histone methyltransferases (HMTs), involved in an intense crosstalk with the DNMTs for regulating chromatin conformation and DNA accessibility [25]. Lysine residues can be mono-, di- and trimethylated, while arginine residues can be monomethylated, or symmetrically or asymmetrically dimethylated [25], with differing outcomes. For instance, a contraction of chromatin is shown when histone H4 is monomethylated on lysine 20 (H4K20me1), or when there are H3K9, K27 or H3R8, with consequent reducing gene expression. Alternatively, transcriptional activation is promoted in the presence of monomethylation of histone H3 on arginine 17 (H3R17me1), or when there are H3R2, R26, H3K4, K36, K79 or H4R3.

Initially, histone methylation was believed to be an irreversible process. This assumption has been retracted by the identification of two evolutionarily conserved families of histone demethylases: LSD (Lys-specific demethylase) and JMJC (Jumonji C) demethylases, which establish demethylation in a different manner. In fact, for guarantying demethylation, LSDs use a flavin adenine dinucleotide (FAD)-dependent amine oxidation reaction, whereas the JMJC family of proteins use a dioxygenase reaction that depends on Fe(II) and α-ketoglutarate [26].

### 2.4. Histone Acetylation

Histone acetylation and deacetylation have a crucial role in the regulation of gene transcription. By removing the positive charge from lysine residues at histone tails, the acetylation promotes the presence of a more relaxed chromatin structure for starting the transcriptional process. The acetylation is catalyzed by the histone acetyltransferases (HATs) Gcn5, P300, CBP and PCAF, and is reverted by histone deacetylases (HDACs), promoting the presence of a closed chromatin with a resulting repression of transcription process. The mammalian HDACs have been classified into four groups: class I, including HDACs 1, 2, 3, and 8; class II, including HDACs 4, 5, 7, 9 (subgroup IIa), 6, and 10 (subgroup IIb); class III, including the Sirtuin family (from SIRT1 to SIRT7); and class IV, including HDAC 11 [27]. 

### 2.5. Histone Phosphorylation

Histone phosphorylation is a post-translational modification, mainly catalyzed by Ataxia-Telangiectasia-mutated (ATM) and Ataxia Telangiectasia and Rad3-related (ATR) enzymes (ATM). This modification affects serine, threonine and tyrosine residues, and it plays an important role in chromatin compaction during meiosis, mitosis and apoptosis. Histone phosphorylation is also linked with transcriptional regulation and gene expression, particularly concerning some genes implicated in the modulation of proliferation and the cell cycle [28]. For instance, the phosphorylation of serine (Ser) 10 and 28 on H3, and serine 32 on H2B has been linked to the activation gene transcription mediated by EGF (epidermal growth factor) [29]. Moreover, histone phosphorylation occurs mostly during DNA damage, on serine (S) 139 of H2A(X) variant histone, also referred to as γH2AX. Involving all phases of the cell cycle, it is involved in the DNA damage response (DDR) pathways, such as non-homologous end joining (NHEJ), homologous recombination (HR) and replication-coupled DNA repair [30].

### 2.6. Histone Ribosylation

Both core histones and the linker histone H1 are subject to mono-ADP-ribosylated modification. Approximately 22 members of the ADP ribosyltransferase (ART) superfamily have been identified, additionally to the members of the sirtuin (SIRT) family, such as SIRT4, as well as SIRT6 and -7, that possess mono-ADP-ribosylation properties. Histone ADP ribosylation is also involved in replication and transcription processes [31]. It has been reported that the lysates of non-dividing cells are predominantly constituted from histones mono-ADP-ribosylated, whereas rapidly proliferating cells are characterized by the presence of poly-ADP-ribosylated histones [32]. 

### 2.7. Histone Ubiquitination and SUMOylation

Histone ubiquitination is determined by the combined action of E1 activating enzymes, E2 conjugation and E3 ligases, which results in the covalent conjugation of ubiquitin (Ub) to a lysine (Lys) residue on proteins or Ub itself, to form different flavors of polyUb chains. In parallel, deubiquitinating enzymes (DUBs) guarantee the removal of these Ub marks. [33]. 

Histone SUMOylation is a recently discovered post-translational modification, thus in respect to the other post-transcriptional modifications, less is known about its effects on chromatin organization and gene expression.

Humans express five SUMO (small ubiquitin-like modifier) paralogs, SUMO-1, -2, -3, -4 and -5 [34]. Shiio and Eisenman first suggested a possible interplay between histone SUMOylation and the other post-transcriptional modifications, such as histone acetylation. For instance, they proved that histone acetylation may facilitate subsequent SUMO conjugation to H4, and they demonstrated, by in vitro studies, the ability of SUMO-1 to link to acetylated H4, and an enhancement of this sumoylation reaction by co-expression of HAT p300 [35].

## 3. Epigenetic Changes in Cellular Senescence

Studies in humans and in vivo models of aging reveal that during aging, the epigenome suffers from a progressive loss in its configuration (which is summarized in Figure 1). Moreover, the chromatin conformation can be perturbed by external stimuli, which consequently influences the expression of genes related to aging and lifespan [36]. This change implicates a profound alteration in the genomic integrity, chromosomal architecture and gene expression patterns [37]. As summarized in Figure 1, there are four interconnected mechanisms that are aging-related. One of the peculiar epigenetic alterations present during aging consists of histone loss and nucleosome repositioning. The loss of histone proteins appears to be a conserved feature of aging, from yeasts to humans. Studies on yeast aging have demonstrated that replicative aging is characterized by a reduced expression of histone proteins, including a decrease by 50% across the whole genome in nucleosome occupancy during replicative aging. This perturbed chromatin composition determines a redistribution of remaining nucleosomes within chromosomes [38,39]. On one hand, it was identified that there is a cluster of genes (i.e., gerontogens) that when overexpressed or mutated increase the lifespan by controlling aging and longevity [40]. On the other hand, these age-dependent alterations produce an increased genomic instability, a loss of specific gene silencing and, in general, increased translation and increased expression of retrotransposons [41]. Furthermore, several studies have demonstrated that this profound change occurs especially in the genes involved in mitochondrial activity and in the modulation of cellular senescence and apoptosis [42,43,44]. To date, the trigger event responsible for histone level reduction, and the consequences of it driving aging remains poorly understood. Experiments in human senescent fibroblasts demonstrated that histone losses were repressed by engineering the expression of telomerase, and as a consequence, DNA damage signals occur for shortened telomeres [45]. Furthermore, it is likely that it is implicated in an inadequate gene expression. For instance, a perturbation in global histone protein levels might influence the global loss of constitutive heterochromatin [46], which in turn could contribute to gene silencing and atypical gene expression.

Another specific epigenetic alteration occurring with aging consists of an aberrant chromatin accessibility (Figure 1). One of the earlier proposed models of aging was the “heterochromatin loss model of aging”, able to explain the modifications in global nuclear building and gene re-expression in the regions involved in the cellular senescence and aging processes [47]. It was demonstrated that this age-related phenomenon, typical for all eukaryotes, may contribute to the shortening or lengthening of the lifespan.

In humans, cells cultured from Hutchinson–Gilford Progeria Syndrome (HGPS) patients manifested a heterochromatin reduction along with heterochromatic mark reduction of H3K27me3 and H3K9me3 [48]. The same defects were identified in skin fibroblasts from the normal control group [49].

Despite a chromatin opening effect on the global losses of histones, senescent cells develop more compact foci of chromatin, termed senescence-associated heterochromatic foci (SAHF) [41]. The formation of SAHF consists of a redistribution of heterochromatin, with a decondensation of heterochromatin, and subsequent formation of foci of heterochromatin in previous euchromatic regions. Additionally, SAHFs are made up of multiple chromatin types, organized in layers with a constitutive heterochromatic nucleus enriched in H3K9me3 and surrounded by a facultative heterochromatic outer ring enriched in H3K27me3 [50]. Furthermore, SAHFs include other typical markers of heterochromatin, such as heterochromatin protein 1 (HP1), hypoacetylated histones, and variant histone macroH2A, and are characterized by their depletion of linker histone H1 [50]. Interestingly, SAHFs have been associated with gene regulation and the irreversibility of cell cycle arrest. In fact, the presence of SAHFs in genes promoting proliferation, such as cyclin A, may subserve to the stable proliferative arrest of senescent cells [51]. Moreover, SAHFs have also been shown to suppress DNA damage signaling [52]. 

Another peculiar age-related variation is the replacement of the canonical histones with histone variants (Figure 1) [51,53]. Histone variants are isoforms of canonical histones that, expressed during the cell cycle, can be incorporated into the chromatin. Their incorporation influences the chromatin structure, consequently modulating crucial cellular processes. Several studies connect histone variants to aging. For instance, an in vitro study of human fibroblasts reported that histone variant, amount and biosynthesis levels, change with aging [54]. 

Histone variant H2A.J is a histone H2A variant linked to aging. The difference between them consists of only five amino acids. It was highlighted that a high expression level of H2A.J is responsible for the expression of inflammatory genes, such as IL1A, IL1B and several interferon-inducible genes that determinate the senescence-associated secretory phenotype (SASP) [55]. To a certain extent, despite cellular senescence can be beneficial, SASP phenotype represents one of the darkest sides of the senescence response because it promotes chronic inflammation that may contribute to and in some cases, aging-associated diseases, such as cancer development.

During ageing some aberrant histone modifications also occur (Figure 1). Among them, the most prominent ones affecting the longevity process are methylation and acetylation of lysine residues. Indeed, during aging, massive modifications have been observed for trimethylation marks on histone H3 lysines 4, 9, 27, and 36 (H3K4me3, H3K9me3, H3K27me3 and H3K36me3, respectively) [40].

Besides the H3K9me3 and H3k27me3, characteristic of SAHFs, the H3K4me3, through the regulation of the expression of aging-related genes, exerts an important activity in the modulation of aging and the human lifespan. Indeed, this kind of histone mark is the most abundant in the close proximity of transcription start sites (TSSs), and it was reported that its high H3K4me3 levels promote aging. By performing RNA interference (RNAi) technology in fertile *C. elegans*, Greer et al. discovered a key regulator of worm lifespan, known as the COMPASS (Complex Proteins Associated with Set1) complex. This complex causes a trimethylation of H3K4 (H3K4me3) and the authors demonstrated that deficiencies in its members extend lifespan [56]. A similar H3K4me3 mark was also identified in Drosophila [57]. H3K27me3 levels appear influenced the lifespan in *C. elegans* models. Indeed, analyzing histone marks associated with repressed chromatin, Maures et al. highlighted the central role of UTX-1, a histone demethylase specific for lysine 27 of histone H3, in the regulation of the worm lifespan. They demonstrated that both heterozygous mutation of *utx-1* or *utx-1* knock-down increase the levels of the H3K27me mark and, consequently, extended the worm lifespan [58,59]. 

Contrary to *C. elegans,* it is plausible that, in Drosophila, H3K27me3 has an impact on the lifespan with an opposite trend. In fact, heterozygous mutations in H3K27 methyltransferase (PRC2) subunits reduce the overall levels of H3K27me3 and increase the lifespan of male drosophila [60]. What is reported demonstrates a variety of epigenetic changes that are species-related. In the context of aging, only few studies were carried out on the epigenome of the animal models. In a study of reprogramming of the naked mole rat cells, in reference to the global histone landscape, it was disclosed that the naked mole rat, well-known for its exceptional longevity, had higher levels of repressive H3K27 methylation marks than mice. In the same experiment, it was also observed that naked mole rats have lower levels of activating H3K27 acetylation marks [61]. 

The pivotal role of histone acetyltransferases and deacetylases in lifespan was also asceratined. For instance, it was demonstrated that SIRT6, by modulating NF-κB signaling, may influence lifespan through the acetylation of H3K9. Indeed, in SIRT6-deficient cells of mice, the hyperacetylation of H3K9, associated with increased apoptosis, and cellular senescence was identified. As demonstrated by an experiment that correlated the catalytic activity of SIRT6 with the deacetylation of H3K9ac with the mitigation of the lifespan of mice [62,63]. H4K16ac falls inside the circle of histones targeting by sirtuin deacetylases [64]. Various studies have shown that H4K16 acetylation is rather reduced during mammal ageing [65,66], but other studies, during aging, associated a decrease in Sir2 with a concomitant increase in the H4K16 acetylation level, resulting in an alteration of transcriptional repression at subtelomeric regions [67]. Next to histone methylation and acetylation, histone phosphorylation and ubiquitination have also been associated with aging. For instance, it was supposed that the histone H2B monoubiquitylation (H2Bub) could indirectly influence ageing by the modulation of H3K4me3 levels [68]. Additionally, an accumulation of H2Bub in heterochromatic regions during cellular aging was identified, accompanied by the increasing level of H3K4me3, H3K79me3 and H4K16ac [69]. 

Interestingly, epigenetic aging modifications are not a consequence of a single alteration, but rather the product of several epigenetic factors that work complementarily. For instance, Rakyan et al. suggested that histone modifications and chromatin remodelers may constitute a baseline for predisposing the DNA to CpG methylation. With increasing age, the cells undergo a global DNA hypomethylation and local DNA hypermethylation, especially at CpG islands near gene-rich regions [70], a pattern which fits the aging model of global heterochromatin deregulation, coupled with a focal increase in repressive modifications. Indeed, it was demonstrated that the global decline of genomic CpG methylation occurs at repetitive regions dispersed throughout the genome, such as SINEs (i.e., Alu elements) and LTR (i.e., HERV-K), that correlate with constitutive heterochromatin. Furthermore, it was highlighted that the hypermethylation usually occurs in genes that control development and differentiation, encode transcription factors or are transcription factor binding sites [71]. Among the latter, two of the most studied transcription factors in aging are Forkhead Box O1 (FOXO) and NF-E2-Related Factor 2 (NRF2), both contributing and responding to the age-related chromatin changes [72]. 

## 4. Epigenetic Changes in Aging-Related Diseases

During ageing, the body is no longer able to maintain homeostasis and becomes more susceptible to stress, disease and injury, as vital bodily functions such as regeneration and reproduction, slowly decline. This condition makes elderly people more susceptible to the onset of age-related pathologies [41].

In this section, we recapitulate the leading epigenetic discoveries in the principal age-related diseases. 

### 4.1. Tumors

In most mammalian species, cancer incidence increases exponentially with advancing age [73], probably because progenitor cells from mature organisms accumulate plenty molecular lesions to evade the homeostatic control of their tissular contexts [74]. The molecular lesions can include genetic (mutations, deletions or translocations) and/or epigenetic alterations. Several epigenetic alterations, such as the local CpG island hypermethylation, and global hypomethylation, collected during aging, play a crucial role in the etiology of complex traits, including cancer [75]. Recently, genome-wide epigenetic studies have revealed particular epigenomic features shared between aging and cancer [76]. Additionally, a growing number of studies have demonstrated that the cellular senescence has a dark side in the cancer, because, on one hand, it may be consider as a safeguard against cancer. On the other hand, it may be involved in cancer aggressiveness [77]. Indeed, senescence represents a potent tumor suppressor mechanism imputable to the permanent cell cycle arrest [78], on the other hand, unlike a static endpoint, there is growing evidence that senescent cells may contribute to oncogenesis, given that the risk of cancer development increases with aging where senescent cells are accumulating [79]. Furthermore, it was demonstrated that senescent human fibroblasts favors the stimulation of premalignant and malignant, rather than normal, epithelial cells to proliferate in culture, and they are able to produce tumors in mice [80]. Moreover, it is likely that the senescence-associated secretory phonotype (SAPS), a phenotype associated with senescence cells, itself encourages tumor cells growth, invasion and metastasis, and tumor vascularization, by secreting inflammatory cytokines, immune modulators and growth factors [81]. Despite this, it remains largely unclear how senescence-related epigenetic modulations contribute to cancer development, several studies have reported a highly significant overlaps between altered DNA methylation in senescent cells and cancer cells. This overlapping is true both for DNA methylation and chromatin changes. In fact, the characteristic scenario of genome-wide DNA methylation decreasing and the presence of site-specific DNA hypermethylation identified in senescent cells are mechanistically linked with genetic instability and the repression of tumor suppressor genes, respectively, typically occurring in cancer cells [70,82].

Both the inducement of cellular senescence and cancer are related to the remodeling of the nuclear envelope and the reorganization of heterochromatin. It is well understood that the association of heterochromatin with the nuclear lamina, guaranteed by lamina-associated domains (LADs), is essential for its stabilization [83]. It also well understood that, during senescence, the modifications in the composition of the nuclear lamina determine changes in epigenetic states and repositioning of LADs, and a loss of peripheral heterochromatin, overlayable to the epigenetic changes occurring in the cancer initiation and progression, supporting the hypothesis that premalignant senescent chromatin changes contribute to oncogenesis [84].

Moreover, some cancers have a specific methylome profile that define a distinct molecular subtype of the cancer. An example is represented by colorectal cancer (CRC) that often affects people over 50 years of age [85]. 

Overexpression of DNMT1 has been detected in several human cancers, including CRC. Zhu et al. reported an increased level in DNMT1 mRNA expression in CRC tissues about twofold when compared with in their corresponding distal normal colorectal mucosa [86]. To this overexpression, an unbalanced methylation pattern in the genome and aberrant methylation in many important tumor suppressor genes is imputable. An increasing expression in DNMT1 was identified in PPARα-deficient mice [87]. Peroxisome proliferator-activated receptor α (PPARα) is a nuclear receptor that serves as a xenobiotic and lipid sensor to regulate energy combustion, lipid homeostasis, and inflammation [88]. In PPARα-deficient mice, it was demonstrated that an increasing expression of DNMT1 leads to decreased levels of p21 and p27, thus promoting cell proliferation and colon carcinogenesis [87]. 

In approximately 15% of malignant colorectal tumors, a hypermethylator phenotype, called the CpG island methylator phenotype (CIMP), was identified, that is seen predominantly in the elderly [89,90]. CIMP is identified in a subset of CRC that happen through an epigenetic instability pathway and that are contradistinguish by vast hypermethylation of promoter CpG island sites, resulting in the inactivation of several tumor suppressor genes or other tumor-related genes. Indeed, many genes, that have been reported to be affected in CIMP, have some crucial roles in the cell controlling; for instance, cyclin-dependent kinase inhibitor 2A (*CDKN2A*), the gene coding for the tumor suppressor p16 involved in the regulation of the cell cycle. Additionally, most CIMPs are characterized by promoter CpG island methylation of the MutL Homolog 1 (*MLH1*), a mismatch repair gene, which defects is associated with the microsatellite instability responsible for approximately 15–20% of all CRC cases [91]. In 8–10% of metastatic colorectal cancer (mCRC), a missense mutation occurring in codon 600 of the gene B-Raf Proto-Oncogene, Serine/Threonine Kinase (*BRAF*), is found, causing a substitution of aminoacidic valine for a glutamic acid (V600E), recognizing it as a negative prognostic factor in light of the fact that the patients carrying the mutation have a median overall survival inferior to 20 months.A strongly association with *MLH1* promoter hypermethylation and the presence of this mutation was reported [92]. Considering that hypermethylation of the *MLH1* promoter, associated with drug resistance, has been found in half of patients with microsatellite instability, the identification of CIMP may acquire value for diagnostic and therapeutic stratification [93].

It was well documented that a chronic inflammation in the bowels represents an increased risk of CRC [94]. This status has also correlated with an aberrant epigenetic regulation. In fact, a methylation analysis of human colonoscopic biopsies of rectal inflammatory mucosa demonstrated a hypermethylation in the promoter region of p16, and associated this epigenetic change with an early stage of the neoplastic progression [95]. 

Aside from DNA methylation, it is growingly approved that the occurrence and development of CRC is influenced by histone modification.

Several studies have reported a positive correlation between the global histone acetylation and histological subtype, cancer recurrence, tumor stage, lymph metastasis, and poor prognosis and survival [96,97,98]. For instance, Karczmarski et al., observed a significant increase in the acetylation of H3K27 in CRC samples, with respect to the normal controls [99]. Ashktorab et al. observed that the transition adenoma–adenocarcinoma is intimately related to the expression of HDAC2 [100]. The consequences of these aberrant histone acetylations in CRC are not well understood. Probably, an upregulation of the acetylation of H3K27 has an effect on the upregulation of the gene *CYR61* (cysteine-rich 61/CCN1), which codes for a secreted matricellular protein that binds directly to various integrin receptors and heparin sulfate proteoglycans to regulate many cellular functions [101,102]. High expression of *CYR61* is observed in colon cancer tissues and has been reported to promote cancer metastasis and cell migration [103,104]. Lingzhu et al. ascribed this upregulation to active CYR61 enhancers, such as the pioneer factor, FOXA1, that, along with CBP, is needed to maintain H3K27ac enrichment at the CYR61 promoter. It was demonstrated that this enrichment is a peculiar characteristic formed in colon cancer, but not in normal colon mucosa [105].

Several reports have also indicated abnormal levels of histone methylation in CRC. For instance, Yokoyama et al., observed a hypermethylation of H3K9me3 especially in invasive regions of colorectal cancer tissues and showed that a positive relation between this epigenetic change and the occurrence of lymph node metastasis [106]. An up-regulation of the methylation status of H3K9 it was also reported to play an important role in human colorectal cancer progression, possibly by promoting collective cell invasion, probably performed by the regulation of the centrosome through the LINC complex, required for cell migration [107]. In addition, Tamagawa et al., demonstrated a positive association between the expression level of H3K37me2 and tumor size and poorer survival rates, supposing this datum as a likely independent prognostic factor for CRC in patients having metachronous liver metastasis [55]. Despite the consequence of this up-regulation is not well understood it is probably that it has an effect on the expression of NOTCH2 [108,109]. 

In gliomas [110], gastric cancer [111], and its precursor intestinal metaplasia [112], and in acute myeloid leukemia (AML) [113] it was also possible to identify molecular subtypes based on DNA methylation profile (DNAm). 

The National Cancer Institute sponsored cooperative groups [114] and a large study of meta-analysis of 3,004 patients with high-grade gliomas [115] confirmed that older age represents a negative prognostic factor for glioblastoma (GBM) survival with a short overall survival of 6-9 months [116]. Regarding the age-related epigenetic, Noushmehr et al., identified a glioma-CpG island methylator phenotype (G-CIMP) in GBMs and asserted that it may represent an age-associated effects on survival in older GBM patients [109]. On the other hand, Garagnani et al., identified some CpG islands of ELOVL Fatty Acid Elongase 2 (*ELOVL2*) hypermethylated in mothers compared with offspring [117]. Interestingly, it was demonstrated that the *ELOVL2* gene encodes for a transmembrane protein involved in the synthesis of long polyunsaturated fatty acids (PUFA), that it is involved in crucial biological functions including modulation of inflammation, maintenance of cell membrane integrity, and energy [118]. Indeed, lipid composition is greatly altered in gliomas with enrichment for free fatty acids and polyunsaturated fatty acids [119]. 

Frequently in cancer, including GBM [120], there is abnormal methylation level at lysine 20 of histone H4 (H4K20me) [121].

Studies of genome-wide disclosed that the gene encoding the histone methyltransferase KMT5B (alias SUV420H1) presents frequently hypermethylated and hypo-hydroxymethylated of DNA in GBM [121,122].

It was reported that a KMT5B epigenetic downregulation in GBM determines an aberrant H4K20 methylation pattern that may cause global transcriptomic changes, promoting tumor growth [121].

In gastric cancer (GC), epigenetic deregulations have identified as an early and advanced-stage occurrences. It was demonstrated that external stimuli and genetic factors, with time, are able to modify gastric epigenetic machinery, by inducing gastritis and ulcer development, and then metaplasia, dysplasia, and tumor development [123]. In GC, focal hypermethylation loci were identified in silencing of tumor suppressor genes. An example is represented by the silencing of oncosuppressor *CDH1* (Cadherin 1), typical of GC, encoding for the adhesion molecule E-cadherin [124,125]. It was identified that the CDH1 hypermethylation occurs in the early step of GC development, and it was attributed to it a significant clinical relevance, in terms of prediction of worse overall and disease-free survival (OS and DFS, respectively) of patients [126].

Together with *CDH1*, the methylation status of gene was to be an encouraging prognostic and diagnostic biomarker, also in GC [127,128,129]. In fact, the methylation of the *CDKN2A* promoter was also identified in gastric pre-cancerous lesions associated with H. pylori and EBV infections, providing evidence of the involvement of methylation status of *CDKN2A* in gastric carcinogenesis [130,131].

In sporadic GC, other relevant alterations of the methylation status are also involved in genes acting in DNA mismatch repair (MMR) pathway [132,133]. For instance, it was reported that the methylation of promoter regions of *MLH1* and *MLH2* genes was associated with the GC onset, the progression, and with the chemoresistance to oxaliplatin [134].

In addition, discoveries of the role of histone modifiers in GC have highlighted the complex epigenetic mechanisms of GC onset and progression and the mutual interchange between DNA methylation and histone modifications. For instance, it was demonstrated a role of the histone deacetylase SIRT1 in GC development, through the inhibition of NF-κB signaling [135]. In various model systems, it was linked the deregulation of H3K27 methyltransferase EZH2 to a promotion of gastric cancer tumorigenesis [136]. Interestingly, Meng et al showed both DNA methylation and histone H3K9 dimethylation deregulation on the promoter of the *CDKN2A* gene [137,138]. 

RUNX3 is a runt-domain transcription factor, frequently inactivated in gastric cancer tissues [139], and is highly related to metastatic outcome [140]. It was reported that its suppression is on account of DNA methylation, via modification of histones, in particular trough G9a histone methyltransferase (HMT) and HDAC1 [138].

Despite, it is not yet fully understood which aging-associated alterations contribute to leukemogenesis, it has ascertained that genomic and epigenomic alterations are main hallmarks of aging [141]. With a comprehensive approach by targeted NGS on a cohort of elderly patients, Silva et al. revealed that AML presents with distinct genetic and epigenetic patterns in the elderly [142]. Furthermore, in young and aged murine hematopoietic stem cells (HSCs), by carrying out a comprehensive study of transcriptome, DNA methylome, and histone modifications, were described global epigenetic changes associated with stem cell aging. The authors demonstrated that the aged cells presented a reduced expression of DNA methyltransferases, and altered positioning of some regulatory histone marks, including H3K4me3, H3K27me3, and H3K36me3 [143]. An analysis of the epigenetic landscape of a HSC-enriched population from young and aged healthy donors, also observed an age-associated reduction in H3K4me1, H3K27ac, and H3K4me3, as well as an altered DNA methylation in aged cells [144]. Additionally, Djeghloul et al., identified in aged HSCs alterations in another heterochromatin mark: H3K9me3 [145]. In particular, by in vitro studies in human and mouse HSCs, they demonstrated a reduced level of SUV39H1, a first lysine-specific HMT activity described, and consequently a reduced level of H3K9me3, with consequent perturbation in heterochromatin function [145]. Finally, several studies have highlighted the pivotal role of histone acetyltransferase p300 in driving malignant transformation [146,147] [67], and its role as a primary driver of the senescent phenotype [148]. 

DNA methylation alterations have also been associated with prognosis in various hematological disorders [149,150]. Several studies have demonstrated that epigenetic modifications can alter the expression of genes that regulate cell cycle progression, cell adhesion and migration, p53 signaling, apoptosis, WNT signaling, cell differentiation, and DNA repair [151,152,153]. 

### 4.2. Cardiovascular Disease

Cardiovascular disease (CVD), prompting to heart failure, and consequently to death, is the principal cause of morbidity and mortality worldwide [154]. Among the risk factors contribute to CVD onset, it is included hypertension, diabetes and obesity [155]. However, the most important determinant of CVD is a person’s age [156], with its prevalence, including atherosclerosis, stroke and myocardial infarction, increasing in the elderly [157].

Several studies suggest that CVD are significantly linked with exposure to chemicals stimuli present in air, food, and water, highlighting the assumption that CVD, as the other pathologies, is a result of a mixture of epigenetic effects and gene alterations [158,159].

Several epigenetic alterations have been linked to cardiovascular diseases and cardiovascular aging; among the others, the DNA methylation, histone methylation and acetylation.

Recently, DNA methylation has been associated with the expression of candidate genes related to coronary heart disease, heart failure, hypertension, and other CVD. Westerman et al., through a genome-wide Bonferroni multiple assay correction, found DNA methylation in three regions, associated with *TNRC6C* (Trinucleotide Repeat Containing Adaptor 6C), *SLC9A1* (Solute Carrier Family 9 Member A1), and *SLC1A5* (Solute Carrier Family 1 Member 5), that were linked with cardiovascular disease risk [160]. Peter J. et al., demonstrated a significant role of SCL1A5 in heart failure. In fact, with respect of the controls, they found a SLC1A5 mRNA levels suppressed in human failing myocardium. Considering that SLC1A5 is crucial for cellular glutamine homeostasis and given the pivotal role of glutamine for cardiac homeostasis, an inhibition of SLC1A5 expression in the myocardium may be responsible to reduce of glutamine uptake, with a consequence alteration of glutamine homeostasis that may contribute to myocardial energy derangements in the heart failure [161]. In epigenome-wide association studies, it was identified thirty-four new DNA methylation sites associated with acute myocardial infarction. Four of them (*CLDND1*: Claudin Domain Containing 1, *AHRR*: Aryl Hydrocarbon Receptor Repressor, *MPO*: Myeloperoxidase, and *PTCD2*: Pentatricopeptide Repeat Domain 2), involved in lipid metabolism and inflammatory diseases, were associated with coronary heart disease [162].

In a cohort study of heart failure, Glezeva et al. detected five genes hypermethylated in the ventricular septal tissues of heart failure patients. Among them, they found promoter hypermethylation of *HEY2* (Hes Related Family BHLH Transcription Factor with YRPW Motif 2) and *MSR1* (Macrophage Scavenger Receptor 1) genes. The first is an important effector in the Notch developmental pathway with a critical role in heart development, whereas, the second is a macrophage-restricted gene responsible for optimized inflammatory response and lipid homeostasis. Furthermore, they identified three genes that showed hypomethylated state. Among them, there is the *CCN2* (Cellular Communication Network Factor 2) gene that is highly expressed in failing hearts, and *MMP2* (Matrix Metallopeptidase 2) gene that is a well-known regulator of collagen turnover and fibrosis, whose serum level results higher in heart tissue of patients at risk for or with established heart failure [163].

Chronic inflammation represents a crucial event because, by inducing cardiomyocyte stress, causes the inception of the pathology. Gentilini et al. connected this condition with unbalanced lipid levels exhibiting during atherosclerosis [164]. For instance, in patients with familial hypercholesterolemia was found a high level of methylation at the promoter region of *ABCA1* (ATP-binding cassette A1), which is required for cholesterol transfer from blood to high-density lipoprotein particles. Furthermore, epigenetic changes were also identified in several genes including *RELA* (RELA Proto-Oncogene, NF-KB Subunit)*, NOS3* (Nitric Oxide Synthase 3)*, KLF4* (KLF Transcription Factor 4) and *APOE* (Apolipoprotein E) that have been linked to progression of atherosclerosis [165]. Indeed, several studies have considered the involvement of HDACs and HATs in atherosclerosis. For instance, HDAC3, HDAC5, and HDAC7 can act as a repressor of the KLF4 expression, as resulted experiments on HUVECs (human umbilical vein endothelial cells) [166]. Therefore, an overexpression of HDACs, typical of the atherosclerosis process [167], may lead to a proatherogenic phenotype [167]. Furthermore, being the atherosclerosis an inflammatory disease [168], it was demonstrated that histone acetylation, by HATs, seems to have a proatherogenic role, that is partially modulated by inflammatory transcriptional pathways [167]. This regulation may be explained by the regulation of NF-kB signaling exerted by p300. p300, upon inflammatory stimulus, interplays with the RELA (p65) subunit of NF-kB in the nucleus to remove repressive p50–HDAC1 interactions [169]. This determines an increased acetylation of both p65 itself and histones in the vicinity of NF-kB binding, driving NF-kB-induced transcription [170]. Additionally, several studies have also highlighted a fundamental role of HATs and HDACs in the regulation of CVDs, such as hypertension, pulmonary hypertension, diabetic cardiomyopathy, coronary artery disease, arrhythmia, and heart failure [171]. This regulation may be exerted not only by controlling inflammatory pathways, but also certain cellular processes including myocyte hypertrophy, apoptosis, fibrosis, and oxidative stress. This latter may be increased by an APOE deficiency [172] probably induced by histone deregulation.

In reference to the role of histone methylation in CVD, Papait et al., suggested the essential role of the histone methyltransferase G9a for maintaining the correct gene expression in normal cardiomyocytes and for driving changes in the expression of genes associated with cardiac hypertrophy [6]. Their research demonstrated that G9a orchestrates crucial epigenetic changes in cardiomyocytes both in physiological and pathological conditions. Indeed, they reported an antihypertrophic manner of G9a in normal heart and a hypertrophy manner of G9a in stressed heart.

This is imputable to the ability of G9a to suppress several classes of genes by demethylating lysine 9 at histone H3, including genes encoding proteins involved in contractility and calcium signaling. Finally, these data may be very useful for developing a novel druggable approach based on epigenetic dysregulation.

### 4.3. Neurodegenerative Diseases 

Alzheimer’s disease and Parkinson’s disease are neurodegenerative diseases associated with epigenetic modifications and closely related to the aging process [173].

### 4.4. Alzheimer’s Disease

Alzheimer’s disease (AD) affects more than 55 million people and is the most common cause of dementia worldwide [174]. Despite several genetic alterations being linked with AD, the majority of AD cases do not have a strong genetic basis and are therefore considered to be a consequence of non-genetic factors. Deepening the epigenetic mechanisms would be a link between environmental and genetic factors, allowing an integration of long-lasting non-genetic inputs on specific genetic backgrounds. Nowadays, a growing number of epigenetic alterations in AD have been reported. For instance, it was described that a predominant risk factor of AD, promoting its onset, is imputable to an accumulation of dysregulated epigenetic mechanisms in aging. Moreover, mutations in different epigenetic regulators have been discovered contributing to impaired learning and memory formation, that are first symptoms of the disease [174,175]. An example is represented by a direct link between DNMT1 alteration and a neurodegenerative disorder of both central and peripheral nervous system. Indeed, DNMT1 mutations were revelled by studies of exome sequencing on HSAN1 (hereditary sensory and autonomic neuropathy) patients. It was demonstrated that the identified mutations are responsible of premature degradation of mutant proteins, a reduction of methyltransferase activity and impairment heterochromatin binding during the G2 cell cycle phase, leading to global hypomethylation and site specific hypermethylation.

Epigenetic dysregulation has also been supposed in Alzheimer’s dementia through reduced histone acetylation.

Despite the discordant results, histone acetylation has been broadly analyzed in relation to transcriptional and memory changes in the aging rodent brain [9]. It is likely that this variation acts by reducing the expression of specific genes implicated in the maintaining of functional synaptic [54]. Growing evidence connects an alteration of histone acetylation homeostasis with the memory loss in AD patients. In normal mice during learning processes, a transient rise in the acetylation of the hippocampal histones was observed, implying that histone acetylation has a crucial role in memory consolidation [176,177,178].

Gjoneska et al., in the p25 transgenic model of AD, not only identified a decreasing in H3K27 acetylation levels at regulatory regions of synaptic plasticity genes, but also identified an increase in H3K27 acetylation levels at regulatory regions of immune response genes [179]. On the other hand, while histone acetylation shows a general decrease in the aged mice, several studies in cellular and animal models of AD have indicated that HDAC inhibitors have a neuroprotective role, sustained memory and synaptic functions [180,181].

Genome-wide studies have emerged as the presence of DNA hypermethylation and histone deacetylation in AD patients, suggesting a general repressed chromatin state and epigenetically reduced plasticity in AD. Generally, as demonstrated in twin-studies, an acceleration of epigenetic age was related to the onset and progression of neurodegenerative diseases [182,183]. In 2009, Mastroeni et al. analyzed DNA methylation in a pair of monozygotic twins discordant for AD. By using immunohistochemical methods to detect 5-methylcytosine (5mc), the authors discovered, in the twin with AD, a significantly decreasing of global DNA methylation within the anterior temporal neocortex and the superior frontal gyrus that was not identified in the neurologically normal, non-demented twin [184]. These analyses may explain the significant synapse loss in the frontal cortex typical of AD [185]. However, since then, other studies using similar techniques have been carried out, but the data were discordant. For instance, Coppieters et al. found the DNA methylation was higher in the middle frontal gyrus of individuals with AD than in the same region of cognitively normal controls [186]. Analogously, Rao et al. confirmed a higher methylation status in the frontal cortex of AD patients [65]. The methylation data have also been discordant for hippocampus; clarifying the epigenetic changes also in this cerebral area would be important given that one of the characteristic trait of AD is cerebral atrophy, and given the role of hippocampus in a memory development [66]. The discrepant data may partly be due to the use of mixed cell populations, leading to deviations across studies. Furthermore, given that the epigenetics has an age-related change, the discrepant data may be imputable to the age differences among the examined samples.

Additionally, several studies reported DNA methylation modifications in specific genes implicated in AD pathology. For instance, in the cerebral area of cerebellum, superior parietal lobe, and inferior temporal lobe of AD patients, Iwata et al, by using pyrosequencing analysis, found statistically significant differences in DNA methylation levels at the promoter region of APP (Amyloid Beta Precursor Protein), MAPT (Microtubule Associated Protein Tau) and GSK3B (Glycogen Synthase Kinase 3 Beta) [187]. Moreover, Wang et al, by mass spectrometry on post-mortem prefrontal cortex samples of AD individuals, found growing DNA methylation levels at the promoter region of *APOE*. Additionally, Yu et al., by comparing the DNA methylation levels in the dorsolateral prefrontal cortex of normal and AD patients, identified 28 gene loci linked with AD pathology. Among there, promoter methylation of *SORL1*, *ABCA7*, *HLA-DRB5*, *SLC2A4* and *BIN1* was positive associated with pathological AD [188]. With respect to DNA methylation, the histone modifications in AD are less studying, with data that are mostly indirect. For instance, Sun et al., suggested that a hyperexpression of HDAC2 contributes to microglial activity and exacerbated inflammation. By experimental animal models, the authors demonstrated that suppressing HDAC2 expression may be a potential therapeutic approach for containing inflammation-induced cognitive deficits [189].

### 4.5. Parkinson’s Disease

Parkinson’s disease (PD) is a neurodegenerative disease with a multifactorial origin. Despite some molecular mechanisms have been demonstrated an involvement in PD development and progression, thus far, PD has a partially obscure molecular profile. In this scenario, the influence of environmental factors on epigenetic machinery modulation acquires a relevant matter for understanding PD pathogenesis. A key protein involved in PD is α-synuclein (α-Syn), encoded by *SNCA* [190]. It was displayed that DNA methylation of *SNCA* has a role in its expression [191]. Indeed, in multiple brain regions of PD patients and, by in vitro studies, it was proved a positive correlation between the hypomethylation in intron 1 of *SNCA* an increase in its expression [191,192]. *DNMT1* is one of the regulators of DNA methylation that appears to play a prominent role in *SNCA* expression. It was proved that the first intron of *SNCA* interacts with *DNMT1*.

In animal models and patient brains, was demonstrated the ability of α-Syn aggregation to sequestrate DNMT1 [193]. This activity was likely ascribed to the DNA hypomethylation of *SNCA* gene and increased SNCA transcription. Indeed, in vitro studies, demonstrated that the hypomethylation of SNCA and increased SNCA expression were mediated by decreased occupancy of DNMT1 in the SNCA promoter region [194]. In peripheral blood mononuclear cells (PBMCs) of PD patients, it was also found a reduced levels of methylation of *SNCA* intron 1. Studies of methylation status of SNCA in leukocytes, demonstrated in healthy individuals a diminished level methylation in SNCA age-related [195], and gender-related. Indeed, among these cohort of patients, it was emerged that men had lower methylation of *SNCA* than women, status that may contribute to the higher incidence of sporadic PD in men [196,197]. While a general reduction of DNA methylation was identified in some brain regions in PD patients, in several brain regions associated with PD, at enhancer regions of the *SNCA* locus, was discovered an enrichment of H3K4me3, and H3K27ac, next to a reduced level of H3K27me3 [197]. The implication of histone modifications in SNCA expression was first described in a patient heterozygous for the *SNCA* p.A53T mutation, and was confirmed by using HDAC inhibitor in an in vitro study [198]. Furthermore, epidemiological studies reported that β-2 adrenergic receptor antagonists raise the incidence of PD. Indeed, β2-adrenoreceptor agonists were discovered to regulate SNCA transcription by H3K27 deacetylation at its promoter, encouraging dopamine neuron health by reducing SNCA expression [199]. Finally, in post-mortem brain samples from PD patients, it was found an increased level of H3K4me3 at the SNCA promoter and this increase was found positive related with higher levels of α-Syn with promotion of neurotoxicity, and cell death [200].

### 4.6. Diabetes

The type 2 Diabetes (T2D) is characterized by chronically elevated blood glucose levels, imputable to insulin resistance in combination with impaired insulin secretion by pancreatic β-cells [201]. It is widely unveiled that aging, a sedentary lifestyle, and obesity contribute to insulin resistance in target tissues including skeletal muscle, liver, and adipose tissue [202]. Nevertheless, pancreatic islet β-cells function decreases after long-term exposure to elevated lipid and glucose levels [203]. Generally, diabetes is a genetic disease but enviromental stimuli carry out crucial roles in its development and progression, by exerting epigenetic changes that modify gene expression. The contribution of epigenetic dysregulation has been evaluated in different models of diabetes by in vitro and in vivo studies. It is well established that the own β cells identity is preserved by controlling DNA methylation. Indeed, on pancreatic β cells of mice deficient in *Dnmt1* it was ascertained a conversion of β cells to α, for a (re)expression of the lineage determination gene *Arx* (aristaless-related homeobox), that is normally methylated and silenced in β cells, and down-methylated and expressed in α cells [204]. In addition, another gene, *PDX-1* (pancreatic and duodenal homeobox 1), essential for β cells differentiation, is also regulated by DNA methylation. In pancreatic β cells of mice, it was demonstrated that diabetes is imputable to pdx-1 silencing. Indeed, in β cells from patients with T2D, DNA methylation was observed meaningfully increased in *PDX-1* promoter and enhancer regions compared with healthy donors and this increase was linked to diminished expression of PDX-1 [205].

Finally, in diabetes it is very important the expression of the *INS 1* (insulin) gene, that is also regulated by DNA methylation. It was reported that high glucose levels induce DNA methylation at *INS 1* promoter [206]. A genome-wide DNA methylation analysis revelled, which others, a differential rate expression of CDKN1A (p21) in human pancreatic islets from T2D and non-diabetic donors. In fact, it was demonstrated that a reduced glucose-stimulated insulin secretion was related to increasing expression and reduced level of *CDKN1A* DNA methylation [207]. Targeting it genetically or pharmacologically may attenuate insulin resistance, and may represent a possible therapeutic approach to treat this metabolic disorder.

### 4.7. Sarcopenia

Sarcopenia is accompanied by a physically inactive lifestyle, loss of mobility, and malnutrition that contribute to impaired quality of life, morbidity, increased health care costs, and mortality. It is an age-related disease characterized by loss of muscle mass, strength, and function [208]. Despite the decline being a consequence of ageing, there is significant variability between individuals in the rate of loss in old age. Some of the inter-variability can be explained by specific genetic factors [209], but much of the remaining variation is unexplained. Given that growing evidence suggesting that epigenetic processes play a prominent role in the development of many complex diseases, it is likely that same processes contribute also to the aetiology of sarcopenia. A number of human studies have compared DNA methylation in muscle tissue from young versus old individuals [210,211,212] and reported differential methylation of genes involved in cytoskeletal function [213,214,215], axon guidance [210,211], muscle contraction [212], cell adhesion [195,196], calcium [216] and mTOR signalling [213]. Moreover, in a retrospective cohort study, by comparing the muscle transcriptome of normal controls and 119 patients affecting by sarcopenia, was identified that in skeletal muscle of sarcopenia patients there is a dysfunctional of mitochondrial bioenergetics, with a down regulation of oxidative phosphorylation genes [217]. An epigenome-wide association study suggested that the same changes in the epigenetic machine may bestow compromised muscle function in later life. Indeed, the authors demonstrated that the differentially methylated CpGs (dmCpGs) were enriched in genes involved in oxidative phosphorylation, voltage-gated calcium channels, and myotube fusion. Finally, by examining the chromatin architecture of the sarcopenia-related dmCpGs, the authors identified that they were contained at EZH2 target genes and at regions of H3K27 trimethylation [218].

### 4.8. Osteoporosis

Osteoporosis (OP) is a systematic skeletal disease characterized by decreased bone mineral density (BMD) and the destruction of bone microstructure, which can lead to increased bone fragility and risk of fracture [217].

In recent years, with the deepening of the research on the pathological mechanism of osteoporosis, the research on epigenetics has made significant inroad. Studies have demonstrated that epigenetic mechanisms are closely related to osteogenesis, bone remodeling, osteogenic differentiation, and other bone metabolism-related processes [218]. Reppe et al. revealed an increasing level of CpG methylation in *SOST* gene promoter region in 27 postmenopausal women with osteoporosis compared to 36 healthy controls [219]. The inhibitory role in bone formation of the glycoprotein sclerostin, encoded by *SOST* gene, was confirmed by Delgado-Calle et al., who in previous study highlighted a hypomethylation in the *SOST* promotor region in human osteocytes, suggesting, as an attracted possibility, targeting sclerostin for the treatment of osteoporosis [220]”.

Moreover, in vitro study on mesenchymal stem cells (MSCs) demonstrated that osteoporosis is tightly related to aging via epigenetic changes [221]. Alvaro del Real et al., by using the Infinium 450K bead array and RNA sequencing, determined the DNA methylation and the transcriptome status of human mesenchymal stem cells (hMSCs) collected by the femoral heads of patients with osteoarthritis (OA) or osteoporotic fractures. Their results disclosed that the epigenome-wide signature of hMSCs from OA patients displays differentially methylated regions in comparison with hMSCs derived from fracture patients. They demonstrated that the differences in “epigenetic brand” are associated with several genes implicated in MSC differentiation and proliferation, such as *RUNX2/OSX* [222].

Bork et al., by using the Human Methylation 27 Bead Chip Microarray, demonstrated different methylation patterns between MSC isolated from young and elderly donors. Moreover, they prepared a long-term MSC culture for comparing to the elderly’s epigenetic profiles. They observed highly significant differences of methylation status at specific CpG sites, despite the methylation patterns being overall preserved during both aging and long-term cell culture. Among others, the CpG islands that were found hypermethylated upon aging included *HOXA* (Homeobox A) and *RUNX2* (Runt-related transcription factor 2), transcription factors involved in osteoblast differentiation [222].

Several studies reported that in osteoporosis the number of clonogenic bone marrow stromal cells (BMSCs) was diminished. This alteration reduced the levels of Tet1 and Tet2 factors. Generally, during osteogenesis, TET1 and TET2 levels are intensified through an increased interaction with the promoter of Osx *(Osterix)* [223,224], a protein that, along with Runx2 and Dlx5, is driving the differentiation of mesenchymal precursor cells into ostetoblasts and eventually osteocytes [225]. Instead, as demonstrated in tet1 and tet2 knockout mice, it was likely that during aging the osteopenic phenotype is ascribable to a hypermethylation in Tet1 and Tet2 that leads osteoporosis by inhibiting the expression of Runx2 [226].

Finally, Wang et al., described a relevant upregulation of H3K27me3 during the process of osteogenesis [221]. Moreover, Ren et al., in bone marrow aspirates from human adults, found an increased level of EZH2 (a component of PRC2 that increases H3k27me3 levels) [227,228], and Jing et al., proved that this increasing is ascribable to an increase level of H3K27me3 on transcriptional sites of RUNX2. These data induced the authors to propose the employment of H3K27me3 inhibitor as a potential approach for osteoporosis disease management [229].

## 5. Conclusions

Here we have depicted an overview of epigenetic mechanisms and have reported the main epigenetic alterations occurring with aging that render cells more prone to the transcriptional modifications implicated in aging-related disease development and progression. Given the reversible nature of epigenetic mechanisms, understanding these alterations will provide promising avenues for therapeutics against age-related decline and diseases. Furthermore, with the advent of high-throughput epigenome mapping technologies, identified the “epigenomic identity card” of each disease and even of each patient will offer a possibility for discovering new diagnostic, predictive, and prognostic, molecular biomarkers that will have a huge impact on patient outcomes.

## Figures and Tables

**Figure 1 cells-12-01163-f001:**
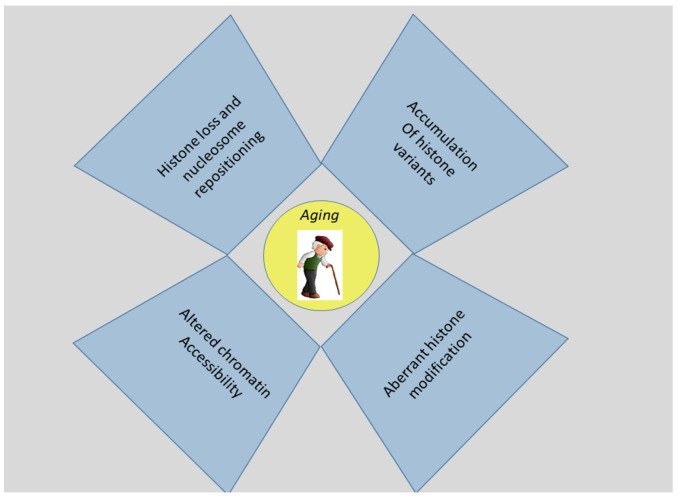
Epigenetic changes in relation to age. The incorporation of different histone variants, altered histone modification and aberrant DNA methylation patterns entail an abnormal chromatin state with the intervention of chromatin modifiers. This abnormal epigenetic scenario induces altered transcriptional patterns and a transcriptional drift within the population.

## Data Availability

Not applicable.

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
