# Peer review of "Epigenetic Mechanisms of Aging and Aging-Associated Diseases"

_cells, 2023, doi:10.3390/cells12081163_

Round 1

Reviewer 1 Report

In this manuscript, Torre summarized the current understanding and progress on epigenetic regulation and aging-related diseases. I recommend that the following points are considered in improving the manuscript prior to publication:

     1.     A high DNMT1 expression level related to CRC was not mentioned in Tumors.

2. The reduced acetylation in AD patients was not introduced in Alzheimer’s disease.

      3.     Line 496, “binding to the Osterix (Osx)should be “binding to the promoter of Osterix (Osx).

The review would be improved with thorough text editing, for example:

4     1.     Line 133, “undergo to” should be “undergo”.

5     2.     Line 177 – 187, the two paragraphs can be merged into one.  

Author Response

REVIEWER #1:

We are grateful to first reviewer for his time reviewing this manuscript, and for the valuable comments. Below we present our detailed responses to first reviewer comments with the comments in black text and our response in red text.

Point 1: A high DNMT1 expression level related to CRC was not mentioned in Tumors.

Response 1: We completed the “Tumors” paragraph as follows:

“Overexpression of DNMT1 has been detected in several human cancers including CRC. Zhu et all, reported an increased level of DNMT1 mRNA expression in CRC tissues about twofold when compared with in their corresponding distal normal colorectal mucosa [73]. To this overexpression is imputable an unbalanced methylation pattern in the genome and aberrant methylation in many important tumor suppressor genes. Increasing expression of DNMT1 was identified in PPARα deficient mice [74]. Peroxisome proliferator-activated receptor α (PPARα) is a nuclear receptor that serves as a xenobiotic and lipid sensor to regulate energy combustion, lipid homeostasis, and inflammation [75]. In PPARα deficient mice, it was demonstrated that an increasing expression of DNMT1 leads to decreased levels of p21 and p27, thus promoting cell proliferation and colon carcinogenesis [74].”

Point 2: The reduced acetylation in AD patients was not introduced in Alzheimer’s disease.

Response 2: We inserted the topic as follows:

Growing evidences connect an impairment of histone acetylation homeostasis with the memory deficit in AD patients. In normal mice during learning processes, it was observed a transiently rise in the acetylation of the hippocampal histones, suggesting that histone acetylation is essential for memory consolidation [Peleg S, Sananbenesi F, Zovoilis A, Burkhardt S, Bahari-Javan S, Agis-Balboa RC, Cota P, Wittnam JL, Gogol-Doering A, Opitz L, Salinas-Riester G, Dettenhofer M, Kang H, Farinelli L, Chen W, Fischer A. Altered histone acetylation is associated with age-dependent memory impairment in mice. Science. 2010 May 7;328(5979):753-6.

Fischer A, Sananbenesi F, Wang X, Dobbin M, Tsai LH. Recovery of learning and memory is associated with chromatin remodelling. Nature. 2007 May 10;447(7141):178-82.

Levenson JM, O'Riordan KJ, Brown KD, Trinh MA, Molfese DL, Sweatt JD. Regulation of histone acetylation during memory formation in the hippocampus. J Biol Chem. 2004 Sep 24;279(39):40545-59].

Gjoneska et al., in the p25 transgenic model of AD, not only found decreased H3K27 acetylation at regulatory regions of synaptic plasticity genes, but also found increased H3K27 acetylation at regulatory regions of immune response genes [Gjoneska E, Pfenning AR, Mathys H, Quon G, Kundaje A, Tsai LH, Kellis M. Conserved epigenomic signals in mice and humans reveal immune basis of Alzheimer's disease. Nature. 2015 Feb 19;518(7539):365-9]. On the other hand, while histone acetylation shows an overall decrease in the aged mice, several studies in cellular and animal models of AD have indicated that HDAC inhibitors have a neuroprotective role by regulating memory and synaptic dysfunctions. In fact, in vitro studies demonstrated the ability of HDAC inhibitors to reverse the acetylation status, decreasing the global histone acetylation and improving the memory deficits [Gräff J, Rei D, Guan JS, Wang WY, Seo J, Hennig KM, Nieland TJ, Fass DM, Kao PF, Kahn M, Su SC, Samiei A, Joseph N, Haggarty SJ, Delalle I, Tsai LH. An epigenetic blockade of cognitive functions in the neurodegenerating brain. Nature. 2012 Feb 29;483(7388):222-6. Walker MP, LaFerla FM, Oddo SS, Brewer GJ. Reversible epigenetic histone modifications and Bdnf expression in neurons with aging and from a mouse model of Alzheimer's disease. Age (Dordr). 2013 Jun;35(3):519-31]. 

Point 3:  Line 133, “undergo to” should be “undergo”.

Response 3: We replaced “undergo to” with “undergo”

Point 4: Line 177 – 187, the two paragraphs can be merged into one.

Response 4: The two paragraphs was merged into one.

Reviewer 2 Report

 This review aims to describe epigenetic modifications on aging-associated diseases.

The title says "epigenetic mechanisms"... but the authors do not explain the epigenetic regulatory mechanisms related to aging or senescence. The most important part of the review consists in listing DNA methylation modifications and some histone modifications (mainly H3K4me3, H3K27ac, H3K27me3) in some diseases, but without explanation of the mechanisms involved in these modifications and only in a few examples explaining the consequences on transcriptional regulation of genes. Overall, this manuscript is a bit frustrating.

More importantly, by leaving out the field of hydroxymethylated DNA and chromatin readers, the authors give an incomplete picture of what this review could be. This diminishes the interest of the reading.

In addition, some of the awkwardness regarding TETs as "DNA methylation eradicators" or the "not purely epigenetic" side of chromatin or noncoding RNA remodeling needs to be corrected.

I do not recommend publication in this form. This review would benefit from a thorough reworking giving more information on the epigenetic mechanisms explaining the relationships between DNA methylation and histone modification and trying to give some interpretation in the description of diseases.

in details :

Line 42 to 45 :« Chromatin remodelers and non-coding RNAs can also participate in the regulation of the chromatin but, because they are not considered purely epigenetic mechanisms, are not included in this review (for a further discussion about these topics see) [8,9].”

The argument should be rephrased : the opposition between histone modification and non-codingRNA as “purely epigenetic” or not is not a good argument. XIST is a purely non-coding RNA important of the X chromosome inactivation. And Phosphoacetylation of H3 during transcriptional activation is not purely epigenetic when considering the definition of “heritable changes” (line36). This review is focused mainly on DNA methylation and histone modifications can be acceptable by itself. It is not required to give inexact idea.

Moreover, several chromatin remodeler are driven to act on chromatin because of histone modifications. Chromatin readers are part of the "epigenetic regulation" because they are involved in the regulation of transcription and replication and, therefore may also be important in aging. The following paragraphs on particular ageing-related diseases illustrate this point : BRAF, line 167; REST, line 363. In addition, several defects in readers of epigenetic marks are involved in aging-related diseases and may be/are examined in the  search of drugs to combat senescence . For example : MeCP2 mediated senescent endothelial cells dysfunction through epigenetic regulation (PMID: 29615114); BRD4 and Senescence Can Be BETter without the SASP (PMID: 27261480); and CBX proteins (PMID: 32870972).

A review with this title should contained a paragraph on this, I supposed.

Line 50 to 70 : DNA methylation is no longer just about methylation or lack of methylation but other Cytosine modifications, which are not just the absence of epigenetic marks, but something else. In 2023, I think it is important to emphasize this in the "context of DNA methylation". This could be broader as "epigenetic modification of DNA". In particular the last paragraph (line 493-498) can benefit from this.

Maybe check for "DNA hydroxymethylation and senescence": the field is vast...

Torres RF et al.. Neural Plast. 2019 Jul 14;2019:5982625. doi: 10.1155/2019/5982625. Writers and Readers of DNA Methylation/Hydroxymethylation in Physiological Aging and Its Impact on Cognitive Function.

Kondo Y. Cancer Res. 2019 Apr 15;79(8):1751-1752. doi: 10.1158/0008-5472. Genome-Epigenome-Senescence: Is TET1 a Caretaker of p53-Injured Lung Cancer Cells?

Filipczak et al. SA. Cancer Res. 2019 Apr 15;79(8):1758-1768. doi: 10.1158/0008-5472. p53-Suppressed Oncogene TET1 Prevents Cellular Aging in Lung Cancer.

Etc… PMID: 22037496, PMID: 30290828, PMID: 32020534, PMID: 22426040...

 Line 59 : DNMT3L is not a a catalytically active protein “it is likely to function as a regulator of methylation at imprinted loci rather than a DNA cytosine methyltransferase” (doi:10.1093/nar/gkf474)., DNMT2 is not DNA methyl-transferase but use the RNA as substrat (doi:10.1101/gad.586710).  DNMT3C is very specific of rodent in testis (DOI: 10.1126/science.aah5143). Please provide better details about this list, maybe by using more recent reviews or even the Wikipedia page.

CpG are not only on CpG islands of the promoter region but also enriched on exons. The conclusion line 68 to 70 needs also better precision : DNA hypomethylation only on the promoter correlates with transcriptional activity, while DNA methylation on the gene bodies also correlates with transcriptional activity (10.1038/nature21373).

To take this idea a step further:: DNA methylation not only regulates the transcriptional initiation activity but also influences RNA alternative splicing regulation through transcriptional elongation (for instance see 10.1101/gr.143503.112 and several other papers).

Line 72 to 98 – histone modifications

Apart from giving a very basic definition of what chromatin is and taking up its regulation by "acetylation" (line 94; of which histone... residues... for what effect?), ...

This section would require an introduction to positive and negative histone methylation which are used in the next section, as well as to chromatin remodelers/modifiers such as SWI/SNF or polycomb. This development could be intended to help understand why a change in DNA methylation affects the recruitment of EZH2/H3K27me3, e.g. line 459... With these explanations, the following descriptions might be more interesting to follow..

Epigenetic changes in aging (line 99 to 144) : aging or cellular senescence ?

Line 122 to 124, please give some examples of histone variants or chromatin regulatory proteins involved in aging modifications and not just references, otherwise it is really frustrating to read and does not bring more interest to the reading of these different reviews.

Line 130 : “HP1” is actually three different proteins alpha, beta, gamma in human and mouse. Please provide the name of the isoform in this study (ref 32).

Again, most of the studies cited in this paragraph are reviews, which are not very new. This give an impression of generalities without real interest.

Tumors (150 to 252)

Line 163, “hypermethylation of the genes involved ….”, -> CIMP are on the CpG islands if I understand well . Maybe the phrase should be : hypermethylation of promoter of genes involved …. This is a general remark for several sentences (among them at least, on lines 213-214, 280, 349, 357, 491) : the authors should refer to the promoters instead of the genes or precise which kind of proximity if it is  possible.

Line 167 : please define what is “BRAF V600E” . Is it BRG1-associated-factor ? Would it be required that the SWI/SNF chromatin remodeler should be introduced in the previous section ? …

Line 226 , define what is the CDKN2A gene.

Line 245 : the code of histone mark (H3K4me3, H3K27me3, and H3K36me3) should be explained in the histone code section. In particular the role of these histone marks are differents.

Line 246-247, it is not clear how histone marks are reduced in aged cells: is there a reduction of histone marks on promoters with these marks or is it the total number of promoters with these marks that is reduced? What are the consequences for gene expression?

Line 249: please clarify in which direction the alteration of H3K9me3 goes?

Cardiovascular (253 to 313)

Line 275 : Did the authors of reference 95 provide the names of the 4 new genes associated with heart disease? 

Line 294, In what direction did the change occur? What types of epigenetic marks have been described on RELA, NOS3, KLF4 and APOE?

Line 303:  “ dimethylation of lysine 9 at histone H3” should be defined in the “histone tail” introduction

Alzheimer (318 to 370)

Line 330 : one or two examples may be interesting to provide to illustrate the reference 102

Line336 : 5-methylcytosine (5mc), should be defined in the “DNA methylation” introduction

Lines 339 to 340: The authors highlighted discordances between studies of DNA methylation in brain but  did not try to suggest ideas to explain it : differences in age, sex, purification of tissues... it is little bit frustrating.

 This paper and other related may be interesting in this field ...
Effect of aging on 5-hydroxymethylcytosine in the mouse hippocampus

Hu Chen  1 , Svetlana Dzitoyeva, Hari Manev (2012,    PMID: 22426040)

Parkinson (371 to 405)

line 379 to 389 : concerning SNCA, the authors pointed out the methylation in intron1 whereas previously in all the review, all DNA methylation reports concerned supposedly the promoter region . This particular case, well-described here, is an example of the importance of DNA methylation not only on promoters of genes. This point may be discussed in the introduction of the DNA methylation.

line 394 ;" an enrichment of H3K4me3, H3K27me3, and H3K27ac at enhancer regions". It is required here to explain why these enrichment could contribute to the SNCA increase in PD, in particular because H3K27me3 is antagonistic to H3K4me3 and K27ac... and somewhat contradictory with the end of the paragraph (line 401-403).

Diabetes (406 to 435)

line 418, Dnmt1 in minus ? Does it mean the sentence referred to mouse study ? it should be indicated.

Osteoporosis (461 to 508)

line 468-474 :  is it clear in this study that the differential 5-mC level is a particular cell type, or it is a mixture of cell types from the whole tissu. In this last case, the change in DNA methylation may be more relevant of the changes in cell population (osteoblast/osteoclast...) than changes in a particular cell. This point could be discussed . In fact also in other studies concerning whole tissue sequencing.

line 494-495 : TET enzymes did not "erase DNA methylation", they catalyse hydroxymethylation of C which constitute another epigenetic mark, also important for transcriptional regulation. hydroxy-meth C is not only the begining of the DNA methylation erasing. (see for example : PMID: 22037496, PMID: 30290828, PMID: 32020534, PMID: 22426040)

Round 2

Reviewer 2 Report

Despite several additions and corrections, some points can still be improved.

I’d like also to point out that some modifications indicated in the cover letter were not highlighted in the reviewed text (for instance line 646-656, corresponding to response 9b)), this did not facilitate the rewiewing.

 Line 57

“DNMT2 and DNMT3L are non-canonical family members, because they do not possess catalytic DNMT activity and represent evolutionary adaptations of original DNMT genes [12].”

I agree DNMT3L “does not possess any catalytic activity” , and it is true that DNMT2 has not “DNMT activity” but it has RNA methyl transferase activity. I think it is important to rephrase this sentence  and cite for example this review :
Albert Jeltsch, et al.
 Mechanism and biological role of Dnmt2 in Nucleic Acid Methylation RNA Biol. 2017; 14(9): 1108–1123. doi: 10.1080/15476286.2016.1191737 

Line 66 : “i---stance”

Line 136 : “phosphor-i-lation”

Chapter3

Point 6: Epigenetic changes in aging (line 174 to 293): aging or cellular senescence?

Response 6: We described the epigenetic changes in relation to age. 

The paragraph 3, is mainly devoted to examples associated with cellular senescence, starting with replicative senescence in yeast (line 184). This is totally different from age-related epigenetic changes that may include cancers (as described in chapter 4 / tumors). Therefore, the title of this chapter 3 should be "epigenetic changes in cellular senescence".

There is no simple epigenetic change related to age since it is related to cells and their specific program to become senescent. Thanks of the details you provide now. However, this new paragraph would benefit from specifying the cell types or animal models from which the results were extracted. For instance, the following sentence is problematic (line 254 to 260),

Indeed, this kind of histone mark is the most abundant in close proximity

of transcription start sites (TSSs), and it was reported that high H3K4me3 levels promote

ageing, contrary to the H3K27me3. It is present along with H3K4me3 mark in a bivalent

domain associated to developmentally important genes. In vitro and in vivo studies have

demonstrated that alteration in the level of H3K27me3 also influences life span, but with

an opposite effect respects to H3K4me3. In fact, level of H3K27me3 were reported to de-

crease with age, establishing transcriptional silencing, which is classically produced by

Polycomb repressive complex-2 (PRC2) and functionally maintained by PRC1 [56].

I guess it is refereing to the ref 39 (line251). Yi SJ, Kim K. New Insights into the Role of Histone Changes in Aging. Int J Mol Sci. 2020 Nov 3;21(21):8241

In this article it is written, I cited “ A further study demonstrated that a reduction in H3K27me3 due to PRC deficiency promotes glycolysis and healthy lifespan [42]. Consistent with this, upregulation of the transcript levels of EZH1 and CBX7/8 and an increase in H3K27me3 were observed in the killifish brain during aging [44]. Additionally, the global level of H3K27me3 was increased in quiescent mouse muscle stem cells during chronological aging [21].”
Their ref42 is related to drosophila.

Thus, the paragraph (lines 254-260) give a misconception of the area that is much more under debate and that may not be so simple the overall increase or decrease of the mark. It depends on the cell types/tissues and perhaps the animal. Perhaps, it helps to cite more precisely the “in vitro and in vivo” studies.

Chapter4

Tumors are indeed age-related diseases as explained by the authors. However, the molecular mechanisms of cell transformation and cell senescence are opposite, leading to the proposal that senescence is a protective mechanism against tumors. In fact, the field is under debate:

Muñoz-Espín et. Al (2014) Cellular senescence: from physiology to pathology. Nat Rev Mol Cell Biol 2014;15:482–96

Gorgoulis et al. ,(2019) Cellular senescence: defining a path forward.Cell 2019;179:813–27

Ines Marin et al . (2023) Cellular Senescence Is Immunogenic and Promotes Antitumor Immunity. Cancer Discov (2023) 13 (2): 410–431. https://doi.org/10.1158/2159-8290.CD-22-0523
Yang et al. (2021) The Paradoxical Role of Cellular Senescence in Cancer. Front. Cell Dev. Biol., 12 Volume 9 - 2021 | https://doi.org/10.3389/fcell.2021.722205

In relation with the previous chapter3 describing mainly epigenetic modification in cellular senescence, it would be interesting to explain this relation between oncogenesis and senescence.

Line 400-402 : “GBM revealed that the gene encoding the histone methyltransferase KMT5B (alias SUV420H1) is frequently hyper- methylated and hypo-hydroxymethylated in this tumor type [107,108].”
Please precise, that the in this context you are talking about the DNA.

Point 11/response 11a.
I apologize for not being clear enough: bone structures are actually dynamic thanks to osteoblasts (builders) and osteoclasts (destroyers) that are in balance. If Delgado's study is to compare an osteoporotic hip fracture to a normal fracture, they are not determining differences between abnormal (osteoporotic) cells and normal cells, but between different types of cells that have invaded the bone. So the question is this: Did Delgado et al sort the cells by FACS, for example, to compare the same cell types and conclude that there was a disruption of DNA methylation in a particular cell type (osteoblast?). Or did they just take the whole tissue without trying to determine which cell type they analyzed? In the latter case, the apparent differential methylation pattern may come from variation in the different cell populations.

Moreover, if Delgado et al, compare hip from patients with osteoporosis versus patients with osteoarthritis the differences can not be only from osteoporisis as explained in the main text. It should be important to point it out, in particular because osteoarthritis is also a age-related disease linked to “too much bone”.
